# Comparison of Lattice Boltzmann and Navier-Stokes for Zonal Turbulence Simulation of Urban Wind Flows

**Marta Camps Santasmasas** [1],*, **Xutong Zhang** [1], **Ben Parslew** [1], **Gregory F. Lane-Serff** [1], **Joshua Millar** [2] **and Alistair Revell** [1]

1 Department of Mechanical, Aerospace and Civil Engineering, School of Engineering, University of Manchester, Manchester M13 9PL, UK; xutong.zhang@manchester.ac.uk (X.Z.); ben.parslew@manchester.ac.uk (B.P.); gregory.f.lane-serff@manchester.ac.uk (G.F.L.-S.); alistair.revell@manchester.ac.uk (A.R.)
2 WSP, Manchester M15 4RP, UK; joshua.millar@wsp.com
* Correspondence: marta.campssantasmasas@manchester.ac.uk

**Abstract:** In modelling turbulent flow around buildings, the computational domain needs to be much larger than the immediate neighbourhood of the building, resulting in computational costs that are excessive for many engineering applications. Two nested models are presented to solve this problem, with an outer domain calculated using a Reynolds Averaged Navier Stokes (RANS) solver in both cases. The inner region is calculated using large eddy simulation (LES) from both a lattice Boltzmann (LB) and a Navier Stokes (NS) based solver. The inner domains use the mean RANS velocity as boundary conditions for the top and the side boundaries and incorporate the RANS turbulence using a synthetic eddy method (SEM) at the inner domain inlet. Both models are tested using an atmospheric boundary layer flow around a rectangular building at $Re_H$ = 47,893, comparing the computational resources spent and validating the results with experimental measurements. The effect of the inlet turbulence, the size of the domain and the cell size are also investigated. Both LB and NS based simulations are able to capture the physics of the flow correctly and show good agreement with the experimental results. Both simulation frameworks were configured to run in a similar computational time, so as to compare the computational resources used. Due to the use of GPU programming, the approach based on LB was estimated to be 25 times cheaper than the NS simulation. Thus these results show that a nested LB-LES solver can run accurate wind flow calculations with consumer level/cloud based computational resources.

**Keywords:** Hybrid RANS LES; Embedded LES; turbulence; urban wind flow; industrial CFD; lattice Boltzmann; GPU

## 1. Introduction

Computational fluid dynamics (CFD) is commonly used in industry to solve a range of wind engineering problems. For many applications, Reynolds Averaged Navier Stokes (RANS) based methods are appropriate as they can generate mean flow field data at low computational cost [1]. However, there are applications for which the time-dependant velocity velocity is important, such as in contaminant dispersion analysis or dynamic wind loading for structural analysis of buildings. In these cases simulations must resolve turbulence, which incurs significant computational cost. It is therefore desirable to explore new options for simulating velocity fluctuations required in engineering applications at significantly reduced computational cost.

Lattice Boltzmann (LB) solvers offer a potential route to resolving turbulent flows with modest computational resources (e.g., Merlier et al. [2], Lenz et al. [3] and Jacob and Sagaut [4]). LB methods contrast to Navier-Stokes (NS) methods by offering a simple and local algorithm, which can be efficiently implemented on massively parallel architectures such as graphic processing units (GPUs). However, traditional LB solvers require high

amounts of memory to store the solved variables and GPU cards have limited available memory. This limits the resolution and domain size that can be modelled on a single node computer. For example Onodera et al. [5] used LB to model a 10 km × 10 km city area at 1 m resolution, requiring 4032 GPUs. Therefore, a major challenge for the application of scale resolving simulation, and in particular for LB, is the amount of the urban environment which must be resolved in order to provide accurate atmospheric flow conditions around the building of interest.

A more efficient approach is to divide the domain into separate regions and use the strengths of different modelling approaches for each region (Figure 1). A large-eddy simulation (LES) can be used to resolve turbulence in a smaller region of interest, while a less computationally expensive RANS model can resolve the surrounding mean flow. Analytic wind profiles estimated from wind measurements are used to define the RANS inlet boundary conditions; while the boudnaries of the LES simulation are the results from the RANS simulation. Mathey and Cokljat [6] used this approach to increase the accuracy of the flow computed in a region of interest around the Ahmed body simplified car geometry; in their approach, the velocity at the inlet of the LES nested region is obtained from the RANS simulation results, the other LES boundaries are independent of the RANS simulation. Embedded LES techniques follow a similar principle of dividing the flow into separate LES and RANS regions, whereby the RANS and LES domains are run concurrently (e.g., Frohlich and Terzi [7]). Jadidi and Bazdidi-Tehrani [8] and Wijesooriya et al. [9] modelled wind flow around a single building using LES. Only the LES inlet boundary was informed by the RANS simulation, i.e., not the side boundaries of the LES domain. Mathey and Cokljat [6] and Jadidi and Bazdidi-Tehrani [8] reported improved results compared to using single RANS models. Wijesooriya et al. [9] reported results from an Embedded LES simulation which where comparable with those from full LES simulation while using 50% of the computational resources. Similarly, Jadidi and Bazdidi-Tehrani [8] reported that the CPU time to complete the Embedded LES simulation is about 50 % lower than that of the full LES simulation. So there is a body of evidence that highlight the benefits of zonal LES methods with NS based solvers.

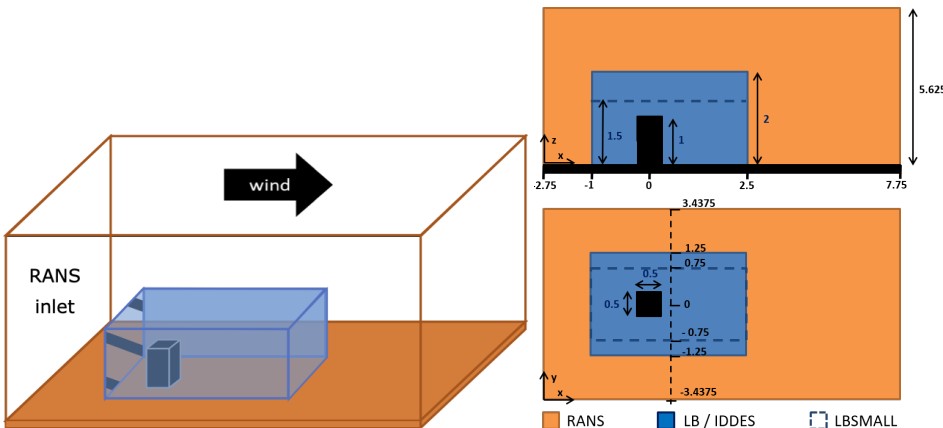

**Figure 1.** Sketch of the domain for the isolated building test case. The orange region is the underlying Reynolds averaged Navier Stokes (RANS) simulation; the lattice Boltzmann (LB) and Navier Stokes (NS) domains simulated in this paper are delimited by the blue region; the LBsmall domain used for the sensitivity test simulation is delimited by blue dashed lines. All the values are in *H* units, where *H* is the height of the building.

LES models require the instantaneous wind velocity to be prescribed at the inlet boundary. However, RANS models only compute mean flow data. Mathey and Cokljat [6], Jadidi and Bazdidi-Tehrani [8] and Wijesooriya et al. [9] addressed this by using a vortex method [10] to generate the instantaneous velocity from RANS mean flow. Millar et al. [11] employed a synthetic eddy method as the inlet boundary condition in an urban wind flow setting, while Buffa et al. [12] applied the original synthetic eddy method (SEM) [13]

to a turbulent flow around a building solved using the LB method; both report good agreement with experimental data. In Buffa et al. [12], the LB method is implemented in a CPU architecture and uses finite differences to approximate velocity and density gradients at the SEM inlet. The present work uses a more recent version of SEM introduced by Skillen et al. [14], due to its improved performance and efficiency compared to original version, modified for use in a GPU LB framework.

Embedded LES approaches commonly use NS based solvers and CPU architectures for both the RANS and LES domains. Another common characteristic is that only the LES inlet is coupled to the RANS results. We present a GPU accelerated lattice Boltzmann LES (LB-LES) simulation nested in a pre-computed RANS domain. The inlet, sides and top boundaries of the nested domain are informed by the RANS results. We hypothesize that the nested LB-LES domain will provide instantaneous velocity results and turbulent statistics that are not present in the pre-computed RANS domain and more accurate time-averaged velocity and turbulent kinetic energy; at a fraction of the cost of running a CPU-based simulation of the same case.

The results of our nested LES simulations are validated against experimental data from Meng and Hibi [15], which is one of the validation benchmark cases in the Guidebook for CFD Predictions of Urban Wind Environment by the Architectural Institute of Japan, and has been previously used to validate LB-LES models (e.g., Han et al. [16]). We performed various simulations to study different aspects of the nested simulation; namely the influence of cell size, the effect of the inlet turbulence and the size of the domain. For comparison, two wall-modelled LES (WMLES) simulations using improved delayed detached eddy simulation (IDDES were undertaken for two mesh configurations. We adjusted the computational resources such that all simulations would take a similar order of time per flow through, and then analysed the accuracy and computational resources required.

This paper demonstrates a novel framework combining a RANS simulation run on CPU with a GPU LB-LES simulation, and provides insight into the accuracy of the approach and the computational efficiencies it can bring. It also highlights the importance of including a synthetic turbulence inlet versus using mean flow data as an inlet for a LES simulation. The new framework is validated against a well-known experimental test case for wind engineering and the efficiency of the CPU-GPU approach is compared against a CPU-only implementation of a Navier-Stokes WMLES of the same case.

The paper is arranged as follows. Section 2 details the LB method and LES turbulence model used in the LB-LES method, while Section 3 describes and validates the method used to generate the instantaneous velocity at the inlet of the LB-LES domain. Section 4 presents the isolated building test case, the algorithm used to run nested simulations and the characteristics of the RANS, WMLES and LB-LES simulations in this study. The results and comparison between these methods are presented in Section 5, before a detailed discussion of the results is given in Section 6. Following this, Section 7 discusses the computational resources used for each simulation and their estimated monetary cost on current cloud hardware. Concluding remarks are provided in Section 8.

## 2. Lattice Boltzmann Flow Solver

The LB equation is based in the Boltzmann equation, which solves for the fluid populations $f$. In the Boltzmann equation, the populations $f(\psi, \mathbf{x}, t)$ represent the probability of a group of fluid particles to move at a certain velocity $\psi$ in a certain position $\mathbf{x}$ and time $t$. The LB equation is a discretisation in time, space and velocity of the Boltzmann equation. The discretisation methods used in this paper leads to the following LB equation:

$$f_\alpha(\mathbf{x_0} + \Delta x, t_0 + \Delta t) - f_\alpha(\mathbf{x_0}, t_0) = \Delta t \Omega(\mathbf{x_0}, t_0) \qquad \alpha = 0 : 18 \tag{1}$$

where $f_\alpha(\mathbf{x_0} + \Delta x, t_0 + \Delta t)$ is the discretised population of the adjacent cell in the $\alpha$ direction for the next time step, $f_\alpha(\mathbf{x_0}, t_0)$ is the population in the current cell and current time step,

$\Delta x$ and $\Delta t$ are the cell size and time step respectively in lattice units and $\Omega(\mathbf{x_0}, t_0)$ is the collision operator. The macroscopic density $\rho$ and velocity $u_i$ of the flow are obtained via

$$\rho = \sum_\alpha f_\alpha \tag{2}$$

$$\rho u_i = \sum_\alpha f_\alpha c_{\alpha i} \qquad i = 1:3 \tag{3}$$

where $c_{\alpha i}$ are the directions the populations can travel from one cell to the next. The number and values of $c_{\alpha i}$ depend on the LB velocity discretisation. The present work uses the common D3Q19 velocity discretization [17], which contains 19 directions in 3 dimensional cells.

The LHS of Equation (1) models the convection of the fluid from one cell to the next, while in the RHS, the collision operator $\Omega$ models the collisions between the particles inside the current cell. We use the Bhatnagar–Gross–Krook (BGK) collision operator, which relaxes the convected population $f_\alpha(\mathbf{x_0}, t_0)$ to its equilibrium value $f^{eq}(\rho, \mathbf{u})$ using the relaxation time $\tau$:

$$\Omega_\alpha = \frac{f^{eq}(\rho, \mathbf{u}) - f_\alpha}{\tau} \tag{4}$$

$$\tau = 3\nu + \frac{1}{2} \tag{5}$$

$$f_\alpha^{eq} = w_\alpha \rho \left(1 + \frac{u_i c_{i\alpha}}{c_s^2} + \frac{(u_i c_{i\alpha})^2}{2c_s^4} - \frac{u_i u_i}{2c_s^2}\right) \tag{6}$$

where $c_s = \frac{1}{\sqrt{3}}$ is the lattice speed of sound in lattice units and $\nu$ is the kinematic viscosity of the fluid in lattice units.

### 2.1. Lattice Units

One convection step of the lattice Boltzmann equations used in this work moves each population from their current cell to the adjacent cell in the $\alpha$ direction. The value of $\Delta x$ and $\Delta t$ in Equation (1) are one cell size and one time step respectively; and have been omitted from the rest of the equations. This assumption, which is common in the standard LB approach, has three well-established consequences: (a) convection is solved without numerical diffusion, since the position of the populations in the previous time step can be traced back to exactly the previous cell in the $\alpha$ direction. (b) The resulting flow fields are in lattice units; for example velocity $u_i$ is measured in cells/time steps and the units of kinematic viscosity $\nu$ are number of cells$^2$/number of time steps. (c) The cells are isotropic in size, which leads to the practical need to find compromise between the size of the domain and the level of resolution required. It is possible to apply mesh refinement in LB simulations (e.g., Rohde et al. [18] and Lagrava et al. [19]) however it complicates the LB algorithm substantially, particularly for the case of GPU implementations. It is our conjecture that the Hybrid NS-LB framework presented in this work can become more efficient than an unstructured LB solver, by leveraging the native advantages of the NS finite volume method implementation.

### 2.2. LES Smagorinsky Turbulence Model

LES for lattice Boltzmann applies the following spatial filtering operation to the populations $f_\alpha$ to obtain the filtered populations $\bar{f}_\alpha$:

$$\bar{f}_\alpha = \int f_\alpha(x) G(x, x') dx' \tag{7}$$

where $G$ is the filtering kernel function. The filtered LB equations become

$$\bar{f}_\alpha(\mathbf{x_0} + \Delta x, t_0 + \Delta t) - \bar{f}_\alpha(\mathbf{x_0}, t_0) = \Delta t \bar{\Omega}(\mathbf{x_0}, t_0) \qquad \alpha = 0:18 \tag{8}$$

Hou et al. [20] implementation of the Smagorinsky turbulence model to LB assumes that the filtered populations will relax towards a filtered equilibrium population that only depends on the filtered macroscopic variables $\bar{u}_i$, $\bar{\rho}$ and that the turbulence closure modifies only the relaxation time $\tau$. Then the filtered BGK collision operator turns into

$$\bar{\Omega}_\alpha = \frac{f^{eq}(\bar{\rho}, \bar{\mathbf{u}}) - \bar{f}_\alpha}{\bar{\tau}} \tag{9}$$

$$\bar{\tau} = \tau + 3\nu_t \tag{10}$$

where the Smagorinsky eddy viscosity $\nu_t$ is

$$\nu_t = C_s \Delta^2 |\bar{\mathbf{S}}| \tag{11}$$

We set the LES filter width $\Delta = 1$ cell and the Smagorinsky constant $C_s = 0.01$. $|\bar{S}| = \sqrt{2\bar{S}_{ij}\bar{S}_{ij}}$ is the intensity of the strain rate of the filtered velocity

$$\bar{S}_{ij} = \frac{1}{2}\left(\frac{\partial \bar{u}_i}{\partial x_j} + \frac{\partial \bar{u}_j}{\partial x_i}\right) \tag{12}$$

The modified relaxation time is [20]:

$$\bar{\tau} = \frac{1}{2\bar{\rho}}\left(\sqrt{\bar{\rho}^2\tau^2 + 18\sqrt{2}\bar{\rho}C_s\Delta^2 Q^{1/2}} - \tau\bar{\rho}\right) + \tau \tag{13}$$

where $Q$ is the second variance of the filtered velocity stress tensor and can be approximated by:

$$Q = \bar{\Pi}_{ij}^{(1)}\bar{\Pi}_{ij}^{(1)} \tag{14}$$

$$\bar{\Pi}_{ij}^{(1)} = \sum_\alpha c_{\alpha_i} c_{\alpha j} \bar{f}_\alpha^{neq} \tag{15}$$

where $\bar{f}_\alpha^{neq} = \bar{f}_\alpha - \bar{f}_\alpha^{eq}$ is the non-equilibrium distribution function. All the operations in Equation (13) are local, thus the implementation of the LES Smagorinsky using Equation (13) does not significantly increase the computational cost of the LB method in GPU. Note that we implemented an LES Smagorinsky model without special near wall treatment.

### 3. Synthetic Turbulence

In order to shorten the development length upstream of the region of interest, we introduce synthetic turbulence fluctuations at the inlet of the domain, superimposed over the upstream mean velocity profile $U_i(\mathbf{x}, t)$, as defined by the reference test case. We achieve this using the SEM based on the version proposed by Skillen et al. [14], which continuously introduces fluctuations at the inlet plane in the form of localised fluctuations in three dimensions, in order to represent turbulent eddies of representative dimensions. In this process a turbulent length scale, $\sigma_i(\mathbf{x}, t)$, must be defined by the user, as well as target values for either the Reynolds stress tensor, $R_{ij}(\mathbf{x}, t)$, or its trace, the turbulence kinetic energy, $k = R_{ii}(\mathbf{x}, t)$. The length scale can generally be approximated directly from the underling RANS calculation.

The SEM algorithm, including the process of generation, scaling and advection of the synthetic eddies through the inlet plane is described in detail in Skillen et al [14]. In the following we focus on how the instantaneous velocity field generated by the SEM is applied to the LB inlet:

1.   Convert SEM velocity field from physical to LB units using the size of the LB cells $\delta x$ and the size of the LB time step $\delta t$. Thus:

$$u_i(\mathbf{x}, t)_{LB} = u_i(\mathbf{x}, t)_{phys}\frac{\delta t}{\delta x} \tag{16}$$

where $u_i(\mathbf{x}, t)_{LB}$ and $u_i(\mathbf{x}, t)_{phys}$ are the SEM velocity field in LB units and physical units respectively. In the following equations, the dimensions $(\mathbf{x}, t)$ has been dropped for clarity.

2. Determine the value of density $\rho$ using the known distribution functions from the LB domain [21].

$$\rho = \frac{\rho_0 + 2\rho_+}{1 - u_n} \tag{17}$$

where $\rho_0$ is the sum of the lattice directions with a particle velocity tangential to the boundary or zero, $\rho_-$ is the sum of the lattice directions streaming from inside the domain and $u_n$ is the macroscopic velocity component perpendicular to the inlet boundary. In the present work, $u_n$ is the streamwise component of the SEM velocity field $u_{xLB}$. The cells in edges and corners of the domain lack some of the components on $\rho_-$, in this case the missing components are streamed from the opposite site of the boundary using periodic boundary conditions.

3. Calculate the missing populations using the regularised LB boundary condition [21] as:

$$f_\alpha = f_\alpha^{eq} + f_{\tilde{\alpha}} - f_{\tilde{\alpha}}^{eq} \tag{18}$$

where $\tilde{\alpha}$ denotes the lattice direction opposite $\alpha$.

4. Recalculate all populations using:

$$f_\alpha = f_\alpha^{eq} + \frac{w_\alpha}{2c_s^4} K_{\alpha ij} \Pi_{ij}^{(1)} \tag{19}$$

where $K_{\alpha ij} = c_{\alpha i} c_{\alpha j} - c_s^2 \delta ij$, where $\delta_{ij}$ is the Kronecker delta.

$$\sigma_i(\mathbf{x}, t) = \frac{R_{ii}^{3/2}(\mathbf{x}, t)}{\epsilon(\mathbf{x}, t)} \tag{20}$$

While the constant cell size in lattice Boltzmann is a drawback compared to the more flexible finite volume implementation, it does provide for a relatively simple optimisation of the SEM algorithm. The standard algorithm has to loop over all cells and all the eddies in order to tally the contribution of each synthetic fluctuation at each grid point; requiring a nested loop which is particularly detrimental to GPU performance. With a structured uniform grid, this search can be replaced with a direct coordinate which substantially improves performance on GPU. While turbulence will continuously reduce in size towards the wall, the minimum prescribed synthetic eddy size, $\sigma_{min}$, will be constrained by the mesh resolution. In this work the length scale was restricted to a minimum of $4\delta x$, while we also chose to limit the maximum eddy size to $H/2$, based on the recommendations in Skillen et al. [14].

Note that all the steps described above are local, they do not require information from the neighbouring cells once streaming is finished. The SEM velocity field is calculated in the CPU part of the LB code. However, the application of this velocity field to the LB inlet is done by the GPU, thus non-local operations would have a detrimental effect on the performance of the code. In future work we will aim to move all of the SEM algorithm to the GPU side of the code. Results of a similar method recently applied to a low-memory implementation of the LB method have recently also been presented in Fan et al. [22].

To assess the present implementation of the SEM, we consider the test case of a turbulent channel flow of height $2H$ at $Re_\tau = 395$, with dimensions $x/H = 10$, $y/H = 2$ and $z/H = 3$. Top and bottom surfaces are set as no-slip walls using halfway bounce back, while the domain is periodic in the spanwise direction. A zero pressure gradient boundary condition is imposed at the outlet, along with a sponge layer that ramps up viscosity in the final period of the domain, to reduce spurious reflections. The SEM is used at the inlet according to the methodology described above.

From Figure 2 we observe that the downstream development of Reynolds shear stress from the LB case is comparable to that from Skillen et al. [14]. While this quantity confirms correct levels of turbulence are generated away from the wall, the skin friction plots in Figure 2b provide close scruty of what is happening directly adjacent to the wall itself. The figure displays the evolution of the dimensionless wall shear stress compared to reference direct numerical simulation (DNS) NS results from Kozuka et al. [23] and results from Skillen et al. [14]. The wall shear stress in the LB simulation is calculated via a 1st order finite difference approximation and there is no special treatment at the wall in the LB method calculation. Despite these simplifications, the predicted $C_f$ is comparable to that obtained by Skillen et al and remains within 2–4% of the periodic DNS at all times. Minor differences are expected due to the absence of a wall model in the LB simulation, which is a deliberate step in the present work to assess capability of a simple implementation. Note that the drop in the predicted value of $C_f$ after $x/H \approx 9$ is due to the sponge layer treatment at the outlet.

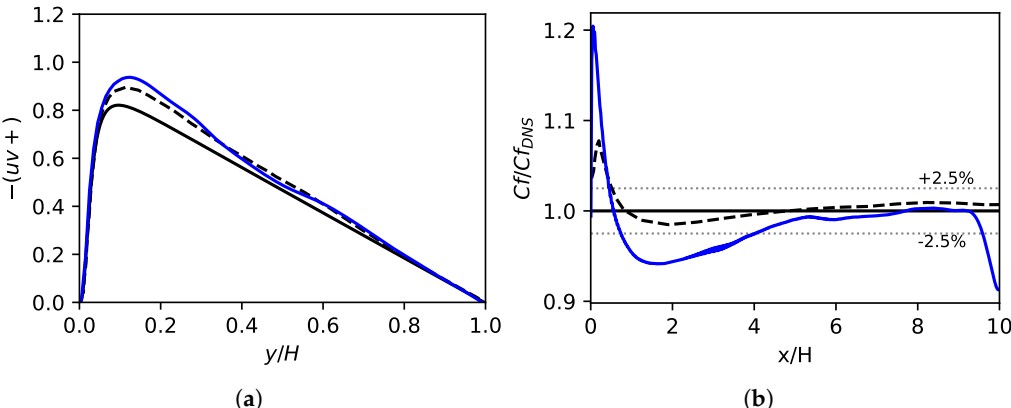

(a)  (b)

**Figure 2.** Application of the synthetic eddy method (SEM) LB solver to the turbulent channel flow at $Re_\tau = 395$. (**a**) Comparison of Reynolds shear stress at x/H = 5 and (**b**) skin friction coefficient along channel wall. Periodic NS direct numerical simulation (DNS) results by Kozuka et al. [23] (——), SEM NS large eddy simulation (LES) results by Skillen et al. [14] (- - -), SEM LB results from implementation described above (——).

## 4. Flow Around an Isolated Building

The present study focuses on the wind tunnel experiment by Meng and Hibi [15]; which consists of flow around a rectangular building at $Re_H = 478{,}000$, where $H$ is the height of the building. This test case is included as test case A in the Validation Benchmark Tests of the Architectural Institute of Japan (https://www.aij.or.jp/jpn/publish/cfdguide/index_e.htm accessed on 30 June 2021 ).

The first step of the nested simulation was to perform a series of RANS simulations of a domain the same size as the wind tunnel experiment. We then extracted mean velocity and turbulence quantities from within the domain to initiate a series of zonal simulations, using both LB and NS based solvers as illustrated in Figure 1. Note that the RANS domain covers the entire volume, i.e., is continuous inside the orange prism. Figure 3 summarises the workflow for the RANS simulation, the zonal LB-LES and zonal NS-LES. An important difference between these two approaches as currently defined is that the NS-LES includes a layer of RANS next to the wall, and is therefore termed a wall-modelled LES (WMLES), whereas the LB-LES solver provides no near-wall modelling as described in the previous section. We performed three additional LB-LES simulations to assess the use of inflow turbulent data in the zonal LB-LES domain inlet, the effect of resolution and the impact of a smaller nested domain and validate the numerical results with the experiment data from Meng and Hibi [15]. Finally, we compare the computational resources used by the WMLES and LB-LES simulations.

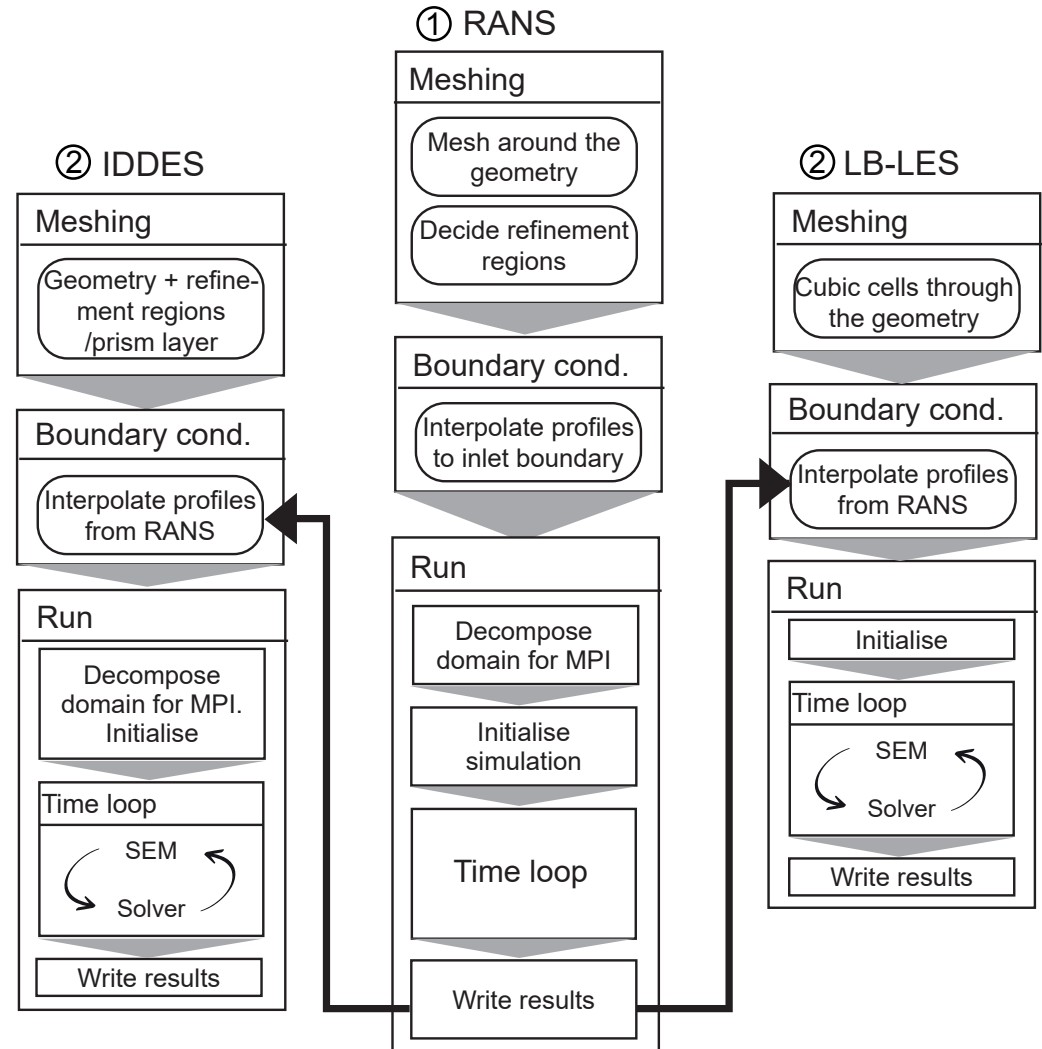

**Figure 3.** Steps to run the zonal simulations in this study; (1) the RANS (centre), (2) either zonal NS wall modelled LES (WMLES) simulations (**left**) or the zonal LB-LES simulations (**right**).

### 4.1. Zonal Navier-Stokes Solver

The RANS and zonal WMLES simulations are run using the is the open source Navier-Stokes based CFD software OpenFOAM [24]; an unstructured finite-volume solver employing a collocated grid-arrangement and second order accuracy discretisation in time and space.

**RANS:** The RANS domain contains 3.3 million cells arranged in a block grid structure (Figure 4). The cell size in the refined region closest to the building is 0.0125 H. Prism layers are applied next to the floor and building surface, yielding a maximum $z^+ = 10.2$. Several additional meshes were generated to assess mesh convergence and the present mesh was deemed sufficient. Note that during this process we employed prior experience to apply block refined regions using octree-based mesh refinement in order to provide a realistic level of efficiency. The boundary condition for the solid surfaces is no-slip with wall modelling, the outlet of the domain is set to fixed pressure and the sides and top are slip boundaries. The inlet velocity and turbulence profiles are fixed to the inlet profiles of the wind tunnel experiment, which provides time averaged turbulent kinetic energy and turbulent dissipation rate profiles. If needed by the turbulence model, the specific dissipation rate $\omega$ is deduced from an equilibrium assumption. We ran the RANS simulation with a selection of turbulence models and compared their results with Meng and Hibi's experiment. The $k - \omega$ SST model returned predictions closest to the experimental data and as such we decided to use those results as boundary conditions for the zonal LES domains.

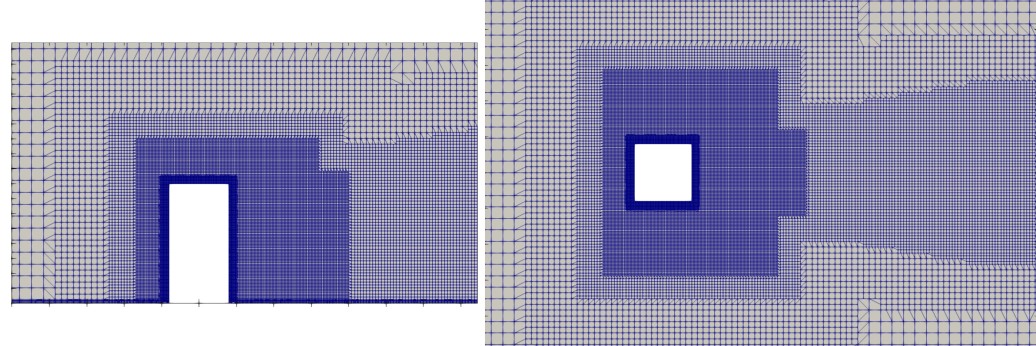

**Figure 4.** Detail of the mesh around the building for the RANS simulation.

**WMLES (IDDES):** $k - \omega$ SST-IDDES is a commonly employed approach for WMLES using NS based solvers, and is adopted here since it employs the same underlying RANS model as the precursor RANS simulation. The WMLES and LB-LES zonal simulations occupy the same space (blue in Figure 1). To avoid confusion, in the following sections we refer to this approach simply as 'IDDES'. The objective is to compare the results, time and computational resources with those using LB-LES. Table 1 summarises the characteristics of the simulations reported in the subsequent sections. The IDDES1 mesh employs regions of refinement around the building to reduce total mesh size (Figure 5a) with the finest refinement region having a cell size of 0.008 H. IDDES2 (Figure 5b) uses a constant cell size equal to LB100 (0.01 H). Both simulations employ near-wall refinement in the form of prism cell layers that reduce cell size in the vicinity of the floor and the building. In these regions the standard IDDES method reverts to the underlying RANS turbulence model, i.e., the $k - \omega$ SST; i.e., the same one used for the precursor RANS simulation. For both IDDES1 and IDDES2, the boundary condition at the solid surfaces is no-slip. The top and side boundaries implement a Dirichlet boundary condition for velocity and turbulent quantities, which are set to the results of the RANS simulation. The outlet implements a zero gradient boundary condition for all the variables. At the inlet, the solver uses SEM to generate the resolved inlet velocity using the RANS mean flow; the modelled turbulence kinetic energy is set to a negligible value to avoid double counting of this quantity. The SEM model and its configuration is the same used for the LB-LES simulations. All IDDES simulations ran until convergence before being time-averaged an additional 815 flow through.

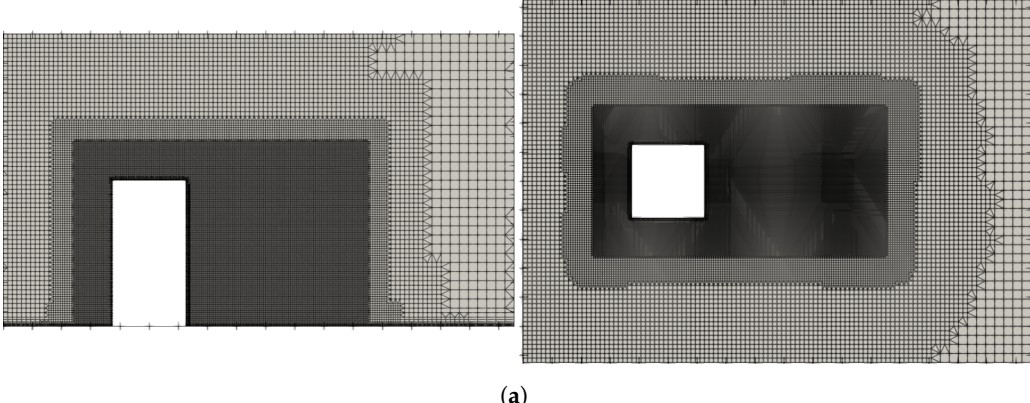

(**a**)

**Figure 5.** *Cont.*

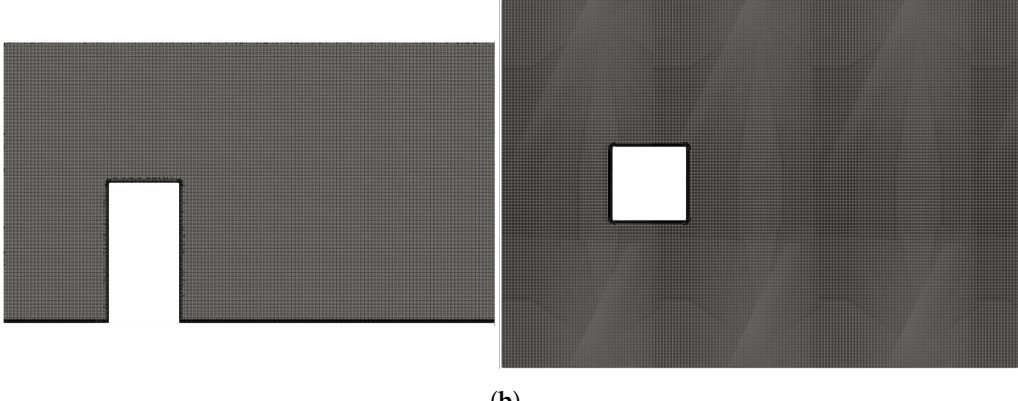

**(b)**

**Figure 5.** (**a**) IDDES1 mesh; (**b**) IDDES2 mesh.

**Table 1.** Numerical parameters for the RANS and IDDES simulations of the isolated building case.

| Name | $\Delta\{x, y, z\}^+$ | Time Step | Cells [$\times 10^6$] | Domain Size/H ($x \times y \times z$) |
|------|------|------|------|------|
| RANS | 10.2 | 0.0001 | 3.3 | $10.5 \times 6.875 \times 5.625$ |
| IDDES1 | 7.2 | 0.0001 | 6.7 | $3.5 \times 2.5 \times 2$ |
| IDDES2 | 1.5 | 0.00005 | 19.1 | $3.5 \times 2.5 \times 2$ |

*4.2. Zonal LB-LES Simulations*

The LB-LES domain size and position is shown in blue in Figure 1. Table 2 summarises the parameters for the zonal LB-LES simulations. The LB-LES mesh is formed by constant size cubic cells. LB100, LBsmall and LBnoSEM have a cell size of $\delta x =$ 0.01 H (Figure 6a) and LB50 has a cell size of $\delta x =$ 0.02 H (Figure 6b). The boundary conditions are equivalent to the ones of the WMLES(IDDES) domains: solid boundaries implement no-slip using half-way bounce back; the top and side boundaries' velocity is set to the RANS results using forced equilibrium [17]; and the outlet implement zero gradient for all the variables via an extrapolated boundary condition, i.e., the populations entering the domain are copied from the cells in the direction of the current population but from the cell previous to the outlet cell. The inlet velocity is obtained and fixed using the RANS results, and the density and populations are obtained using the regularised boundary condition [21]. We tested two different approaches to defining the inlet velocity for this method:

- LBnoSEM: The inlet velocity is set to the RANS velocity. The values of the RANS turbulence variables are disregarded; thus the LB-LES simulation contains only the turbulence generated inside the LB-LES domain.
- LB100, LB50, LBsmall: The solver uses SEM (Section 3) to generate the resolved inlet velocity at each time step.

**Table 2.** Numerical parameters for the zonal LB-LES simulations. $H$ is the height of the building.

| Name | Cell Size/H | Time Step [s] | Domain Size/H ($x \times y \times z$) | Inlet | Focus on |
|------|------|------|------|------|------|
| LB100 | 0.01 | 0.00001 | $3.5 \times 2.5 \times 2$ | SEM | base case |
| LB50 | 0.02 | 0.00004 | $3.5 \times 2.0 \times 2$ | SEM | lower resolution |
| LBsmall | 0.01 | 0.00001 | $3.5 \times 1.5 \times 1.5$ | SEM | smaller domain |
| LBnoSEM | 0.01 | 0.00001 | $3.5 \times 2.0 \times 2$ | RANS | inlet turbulence |

The time averaged velocity used by SEM is set to the RANS velocity, the Reynolds stresses are approximated using the Boussinesq eddy viscosity assumption using the RANS velocity and turbulent kinetic energy (TKE). Finally, the eddy size is calculated using the generated Reynolds stresses and the turbulent energy dissipation rate $\epsilon = k\, 0.09\, \omega$, where $\omega$ is the specific dissipation rate from RANS. The maximum synthetic eddy size $\sigma_{max} = H/8$

and the minimum eddy size is $\sigma_{min} = 4\,\delta x$, where $\delta x$ is the cell size. The SEM length scale calculated from the approximated RANS Reynolds stresses presents only a small portion of the inlet of the domain with eddy sizes between $4\delta x$ and $H/8$, and so this expression yields a constant value of $H/8$ across the inlet, except in the region close to the ground. We chose to limit the eddy size to $H/8$ based on prior experience for turbulent boundary layers and channel flows. Linear interpolation is used to transfer the RANS results to the corresponding LB-LES mesh. All the LB-LES simulations ran until convergence before being time-averaged for and additional period of 1685 flow through.

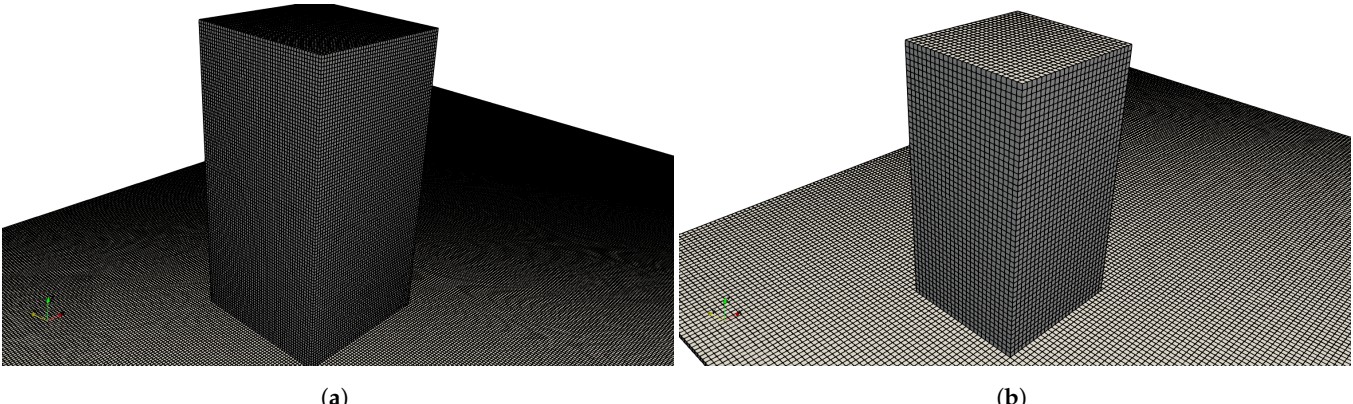

(**a**)                           (**b**)

**Figure 6.** Detail of the mesh around the building for (**a**) the LB100 simulation and (**b**) the LB50 simulation.

## 5. Results

In this section we will provide a summary of the main observations from the simulations described in the previous section. In order to provide an example of the flow configuration, Figure 7 shows the instantaneous LB-LES velocity on the building's centre-plane overlaid on the RANS velocity. This data is accompanied by iso-Q contours in the LB-LES domain and velocity streamlines in the RANS domain, which display the 3D structure of the flow. Fluctuations are visible in the figure which have been generated using RANS velocity and Reynolds stresses. It is also important to note that turbulent structures appear upstream of the building, i.e., not only in the building wake. Finally, note that turbulent structures can also be observed next to the LB-LES top boundary even if that boundary is set to the RANS velocity without SEM.

Figure 8 displays the time-averaged flow pattern in the wake of the building for each LB-LES simulation in Table 2, each zonal IDDES simulation in Table 1 and for the underlying RANS simulation. All cases present the frontal vortex at the front of the building and the main recirculation downstream the building. The main recirculation reattachment length for LB50, LBnoSEM and RANS is between 1.5 and 2. The reattachment length for LBsmall is the longest, between 2 and 2.5, and the length for LB100, IDDES1 and IDDES2 are between 1 and 1.5. All cases reproduce the recirculations above the roof and near the ground on the sides of the building. However, the flow on the roof for LBnoSEM and RANS does not reattach to the building surface and the recirculation on the side is wider than in the other cases. Regarding the overall shape of the wake, in the vertical plane, all simulations present a main recirculation centred close to the roof and a smaller vortex attached to the corner between the building and the ground; the horizontal plane at pedestrian height presents another vortex covering half the width of the building. The main differences in the shape of the wake manifest in LBsmall and LBnoSEM. The main vertical recirculation in LBsmall is longer than in the other simulations and its centre is further away from the building and at a lower height; the recirculation at pedestrian height is also longer and its centre is displaced away from the building and slightly to the side, so that the centre of the recirculation is aligned with the side of the building. The centre of the vertical recirculation of LBnoSEM is aligned with the roof of the building, which makes it the highest position of all the simulations; the centre of the recirculation at pedestrian height is also situated

next to the building and thus it presents the widest wake of all the simulations. The RANS streamlines do not present the frontal vortex and smaller vortex attached to the corner between the building and the ground.

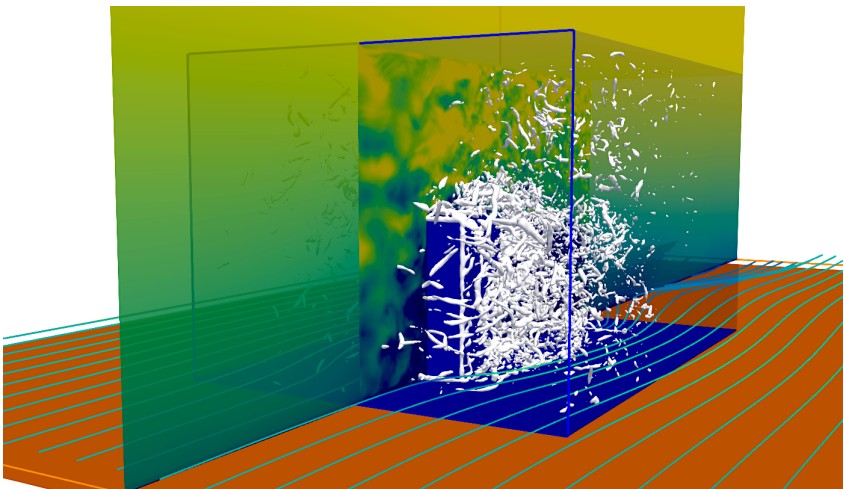

**Figure 7.** Global view of the RANS and embedded LB-LES results. The orange floor corresponds to the RANS domain, blue is the LB-LES domain; the outline of the SEM interface between RANS and LB-LES is blue. RANS results shown are: streamlines coloured by velocity magnitude at $z = 0.00625$ H together with a contour plane of RANS velocity magnitude. The LB-LES results shown are: contour plane of instantaneous velocity magnitude and iso-Q contours at $Q = 1.2 \times 10^5$. The colour scale for velocity magnitude is $v_0 = 0$ in dark blue, $v_{max} = 6$ m/s in yellow.

Figure 9 provides a comparison of TKE contours for each of the simulations and the experimental data. The TKE contours for the experimental data have been generated using the experimental points in Figure 10. For the RANS simulation, TKE is the time averaged TKE modelled by the $k - \omega$ SST turbulence model; for the IDDES simulations TKE is the sum of the resolved turbulence and the time averaged TKE from the turbulence model; for the LB-LES simulations TKE is the sum of the resolved turbulence and the modelled subgrid turbulence.

$$\text{TKE} = \frac{1}{2}R_{ii}^2 + \text{TKE}_{mod} \tag{21}$$

where $\text{TKE}_{mod}$ is approximated using Yoshizawa's model as $\text{TKE}_{mod} = 2C_I\Delta^2|\bar{S}|^2$, with $C_I = 0.01$ [25].

For all cases, an increase in TKE is observed around the roof, the sides and downstream of the building, as is expected for this flow. However, the shape of the wake and the location of the maximum turbulence levels differ from case to case, corresponding to observations made for the mean velocity. LB100, LB50 and IDDES2 present the TKE levels closest to the experimental data. LB100 and LB50 present a similar wake shape, the main difference being the elevated levels of TKE over the roof and in front of the building for LB100. LBsmall presents a narrow region of high TKE downstream of the building, roughly aligned with the roof; near the floor the high TKE region is the shortest of all the 4 cases. In the case of LBnoSEM, turbulence is observed to be low until the wake, with turbulence reaching a maximum downstream of the building. Values over the roof and the front and sides at pedestrian height are substantially lower than those observed with LB100, LB50 and LBsmall, since without the structures introduced by the SEM, resolved turbulence is low and remains as such until the flow becomes fully separated in the wake. Unsurprisingly, RANS results present even lower levels of turbulence, both over and downstream of the building, despite erroneously indicating increased turbulence levels in front of the building, potentially as a consequence of the flow stagnation. It is

also important to note that, away from the building, the vertical plots for the experiment present a nearly constant background TKE level of 0.03–0.06. In the LBnoSEM and IDDES1 results, the background TKE value is in the range 0–0.03. In LB100, LBSEM50 and LBsmall the range is 0.03–0.06 as with the experiment. However, TKE is reduced to 0–0.03 in the region close to the top boundary of the domain. IDDES2 presents a similar pattern with the 0–0.03 region occupying also a significant part of the upstream region and the region on top of the building.

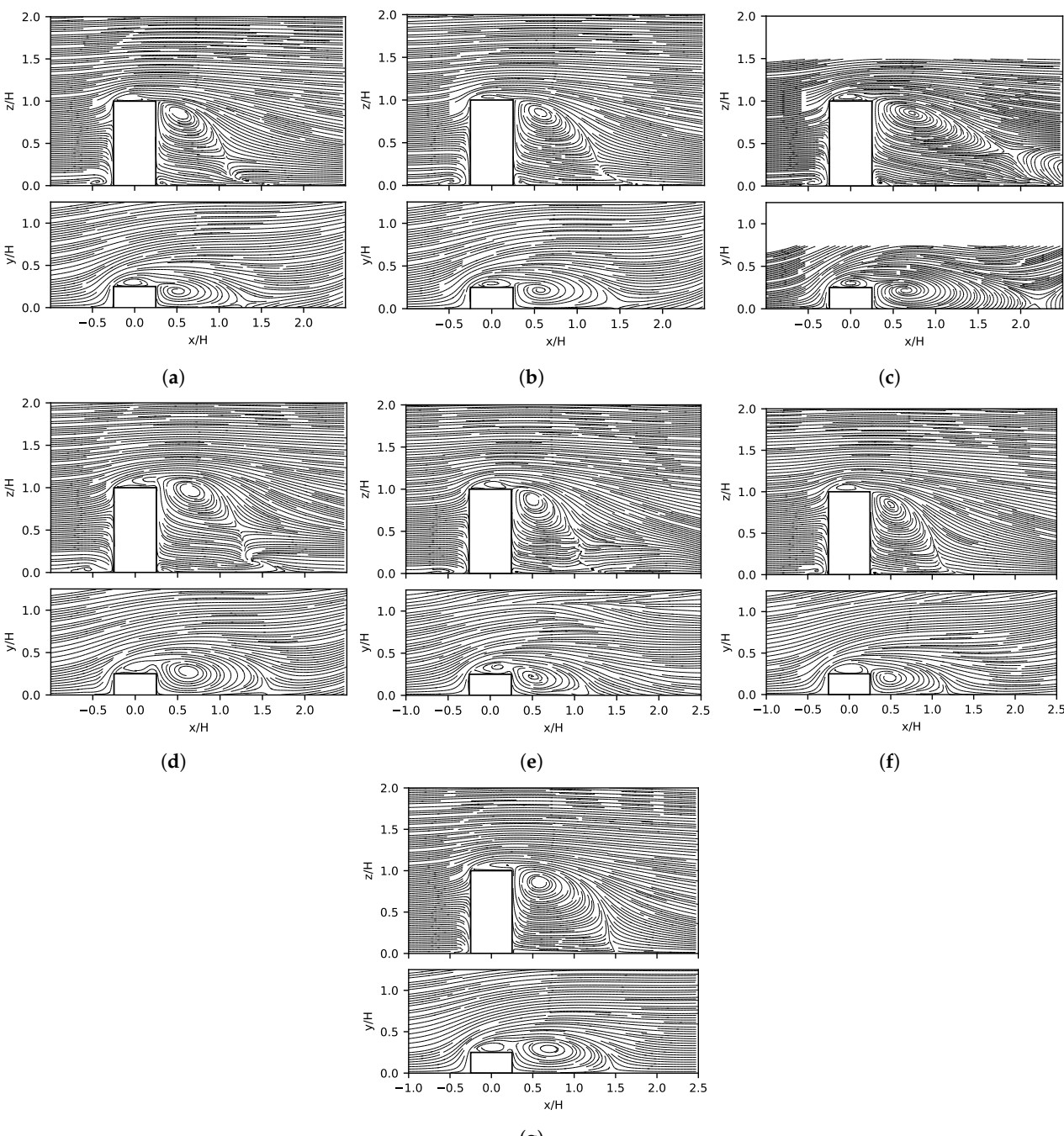

**Figure 8.** Streamlines of time-averaged velocity in the vertical plane ($y/H = 0$) and at pedestrian height ($z/H = 0.0625$) for each of the cases tested. (**a**) LB100. (**b**) LB50. (**c**) LBsmall. (**d**) LBnoSEM. (**e**) IDDES1. (**f**) IDDES2. (**g**) RANS.

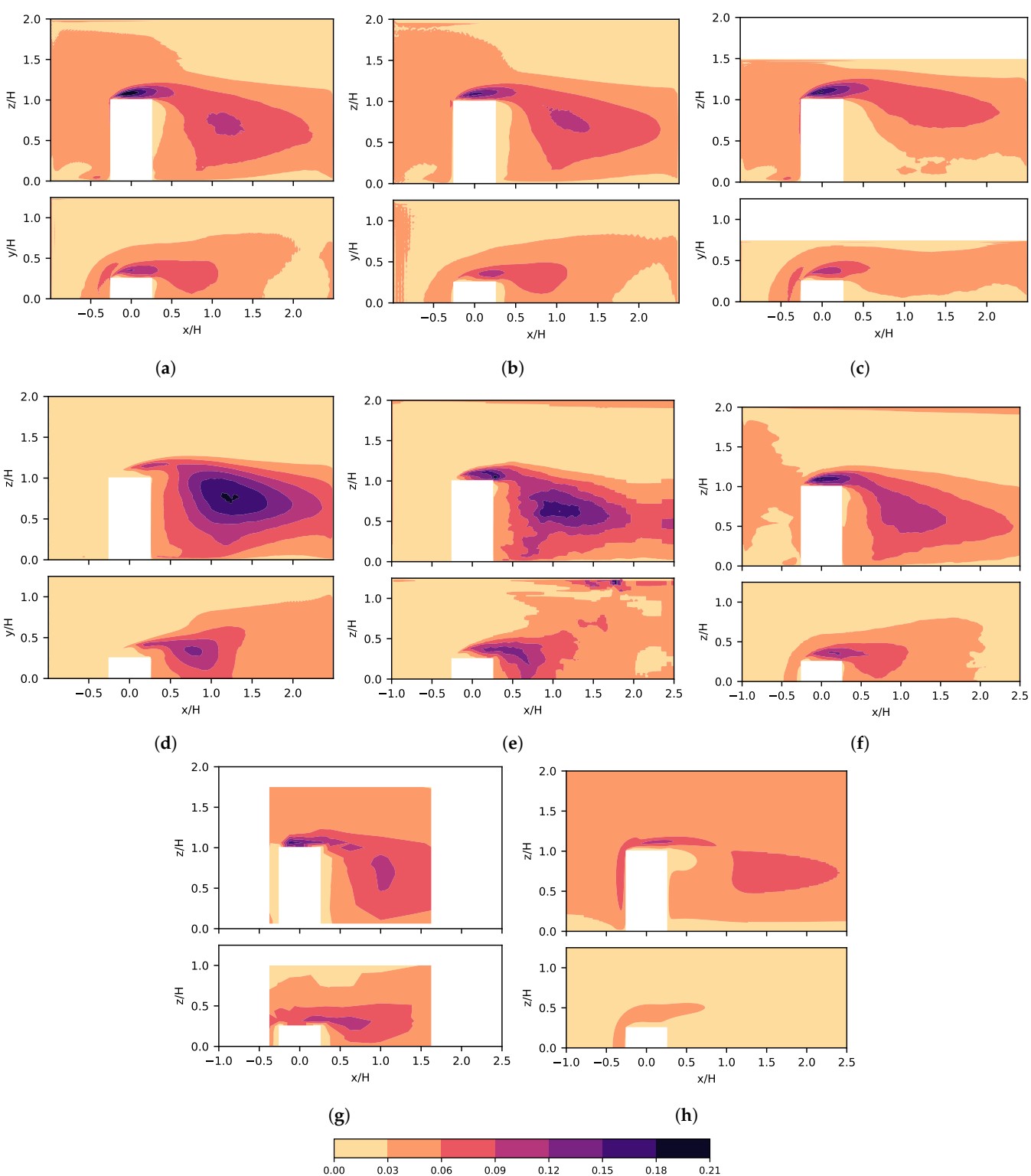

**Figure 9.** Normalised turbulent kinetic energy $TKE/U_b^2$ in the vertical plane ($y/H = 0$) and at pedestrian height ($z/H = 0.0625$) for each of the cases tested. (**a**) LB100. (**b**) LB50. (**c**) LBsmall. (**d**) LBnoSEM. (**e**) IDDES1. (**f**) IDDES2. (**g**) Experimental results [15]. (**h**) RANS.

Further scruity of values is provided in the form of line plots in Figures 10 and 11. Experimental data is available at 3 upstream locations $x/H = -0.375, -0.25, -0.125$, the centre of the building $x/H = 0$ and three downstream locations $x/H = 0.25, 0.375, 0.625$. These locations are sampled through the centreline of the domain $y/H = 0$ and across the

spanwise dimension at $z/H = 0.0625$, equivalent to pedestrian height. Figure 10 displays the mean streamwise velocity for the most accurate of each group of results, i.e., LB100 and IDDES2, together with the RANS simulation and experimental data. As expected, the three simulations match the experimental results in the majority of the domain, the main discrepancies being in the near-wall region of the building. In the vertical plane, IDDES2 and LB100 results follow the experimental results closely; RANS results under predict the velocity over the building, in accordance with the larger re-circulation region observed in Figure 8g. The main discrepancies at pedestrian height are in the wake region near the building. There, RANS results under predict the velocity, whereas LB100 and IDDES2 are indistinguishable and follow the experimental data closely.

Figure 11 provides plots of *TKE* as well as the root mean squared (rms) of the resolved streamwise velocity $< \bar{u}' >$, spanwise velocity $< \bar{v}' >$ and vertical velocity $< \bar{w}' >$, compared with the experimental data. As with velocity, the simulation results match the experimental data in the far field and the main discrepancies are close to the building. The RANS simulation consistently under predicts *TKE* near the building and tends towards the experimental values in the far field, which corroborates the observations in Figure 9. LB100 and IDDES2 are able to capture the resolved scales of turbulence correctly, indicating a close overall agreement with the experimental data.

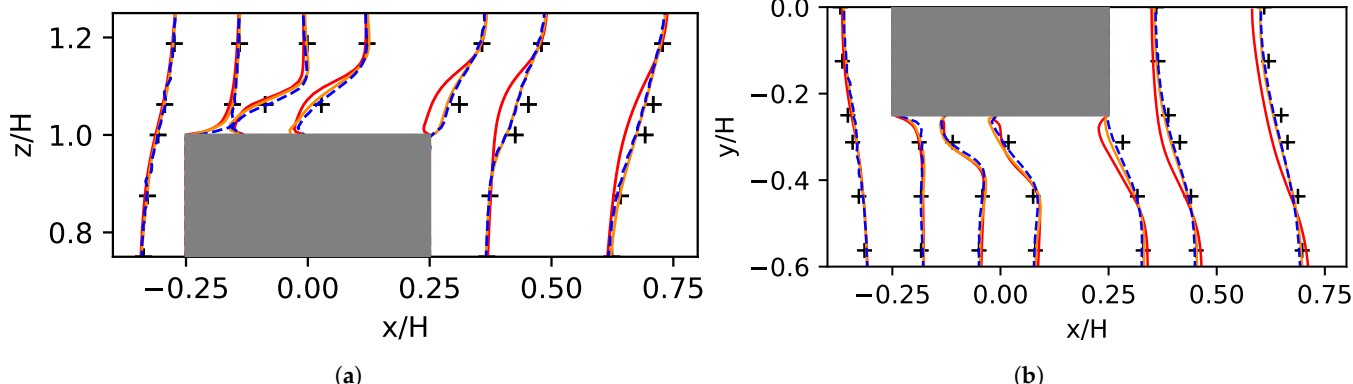

(**a**)        (**b**)

**Figure 10.** Time averaged streamwise velocity $U/U_{ref}$ on (**a**) centreline vertical plane and (**b**) pedestrian height. Meng and Hibi [15] experimental data (+), $k - \omega$ SST RANS (——), LB100 (- - -) and IDDES2(——). (**a**) $y/H = 0$. (**b**) $z/H = 0.0625$.

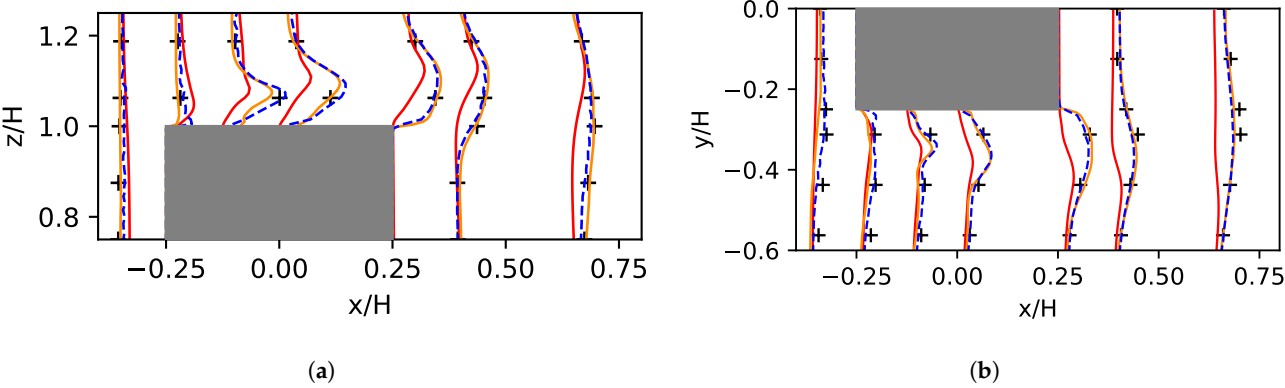

(**a**)        (**b**)

**Figure 11.** *Cont*.

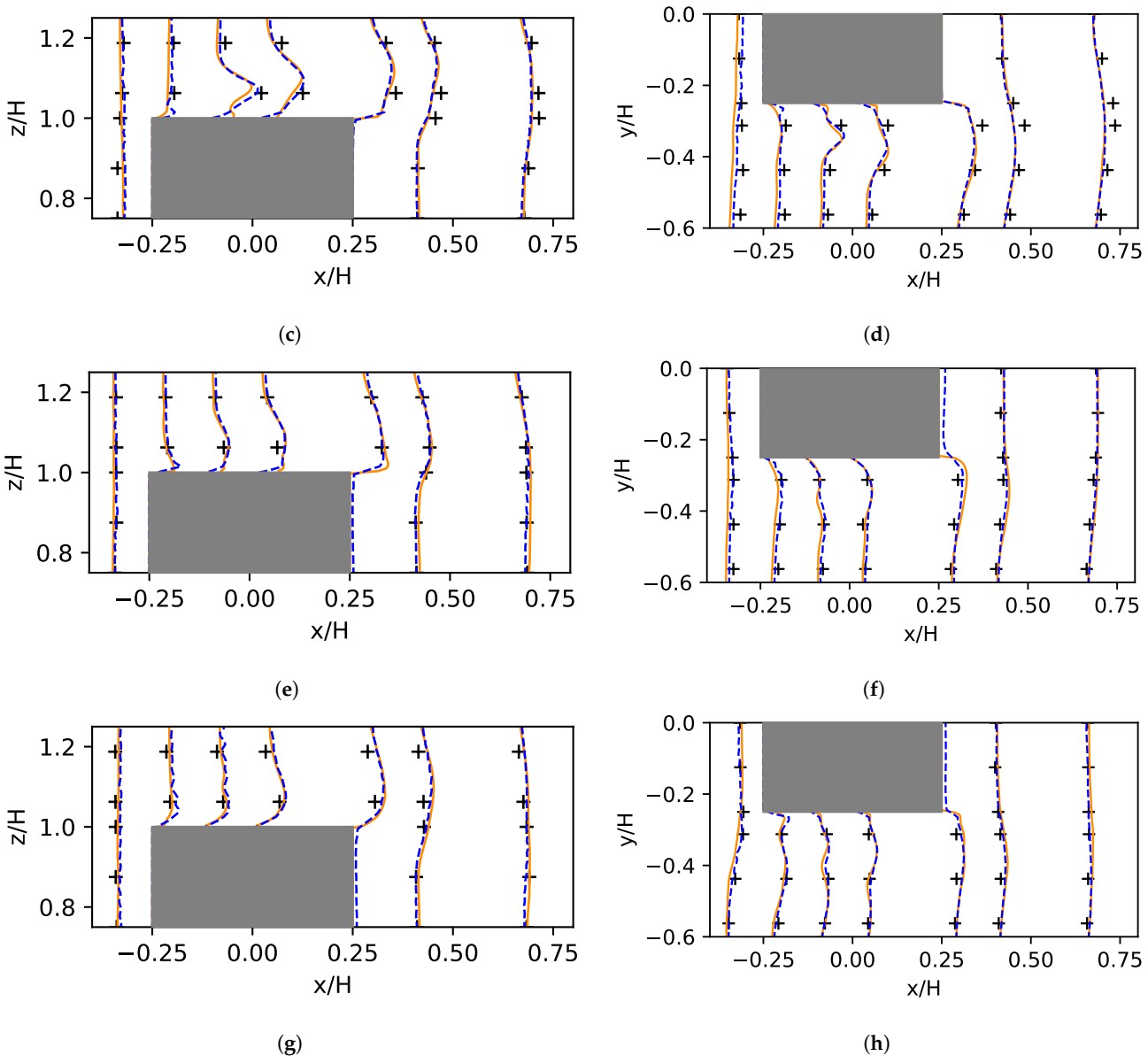

(c)                                                   (d)

(e)                                                   (f)

(g)                                                   (h)

**Figure 11.** Normalised TKE $TKE/U_b^2$ and root mean squared of the fluctuating velocity $< \bar{u}' >$ $/U_b^2$, $< \bar{v}' > /U_b^2$, $< \bar{w}' > /U_b^2$ on the vertical plane $y/H = 0$ (left) and the horizontal plane $z/H = 0.0625$ (right). Meng and Hibi experimental data (+), IDDES2 (——), LB100 (- - -) and RANS (——). (**a**) $TKE/U_b^2$ on $y/H = 0$. (**b**) $TKE/U_b^2$ on $z/H = 0.0625$. (**c**) $< \bar{u}' > /U_b^2$ on $y/H = 0$. (**d**) $< \bar{u}' > /U_b^2$ on $z/H = 0.0625$. (**e**) $< \bar{v}' > /U_b^2$ on $y/H = 0$. (**f**) $< \bar{v}' > /U_b^2$ on $z/H = 0.0625$. (**g**) $< \bar{w}' > /U_b^2$ on $y/H = 0$. (**h**) $< \bar{w}' > /U_b^2$ on $z/H = 0.0625$.

## 6. Discussion

The LB100 and IDDES2 results accurately predict the flow pattern and turbulence levels from the wind tunnel experiments, showing a significant improvement from the underlying RANS results.

The introduction of SEM at the LB-LES inlet is key to the accuracy of the LB-LES results. The SEM simulations accurately reproduce the free stream turbulence levels present on the wind tunnel experiment and RANS simulation. LBnoSEM lacks the free stream turbulence data incorporated into its inlet, which leads to an under prediction of the free stream turbulence and an over prediction of the TKE in the building's wake. Lower values of background turbulence also lead to reduced turbulent mixing, and thus a wider and taller wake than the cases with SEM. It is interesting to note that while IDDES1 also implements

SEM at the inlet, the background turbulence level is observed to be comparable to the LBnoSEM simulation. This is due to the use of a substantial mesh refinement between the inlet and the start of the building, whereby turbulence introduced by SEM is subsequently dissipated by the relatively coarse mesh upstream of the building. Such meshes are commonly used in the industrial use of synthetic turbulence and IDDES, and as such these results are instructive to future use; it is only worth introducing synthetic turbulence if suitable mesh resolution is employed between the inlet and the start of the region of interest. In contrast, the IDDES2 case is able to sustain the inlet fluctuations from the start of the domain to the building. However, even though the IDDES2 mesh has the same cell count as the LB100 simulation, greater dissipation is observed in former case than with the latter; resulting in notably lower levels of TKE. While it is difficult to make direct comparison between the two methods and solvers, higher levels numerical dissipation are expected in the IDDES finite volume formulation compared to the LB-LES formulation, due to the modelling and the numerical treatment.

A small drop in turbulence along the top boundary is observed in the LB100, LB50 and LBsmall, which increases with the distance to the inlet and is not present in the RANS and experimental data. At present, boundary condition on the top and sides boundaries of the LB-LES simulations are implmented as simple, forced equilibrium and the velocity is imposed directly from the pre-cursor RANS simulation. Flow is therefore set to take a constant value on this boundary and an adaptation region is expected here, in which the the velocity adjusts from a constant value to become fully unsteady. This region is imperceptible in the instantaneous velocity plot (Figure 7), probably due to the nature of the forced equilibrium boundary condition (i.e., the LB equations in equilibrium reproduce the inviscid Navier-Stokes equations [17]). Further improvements may be possible in future by implementing SEM to act on top and side boundaries also. In particular, the effect of these relatively simply top and side boundary conditions has a visible influence on the LBsmall results. Where there is a need to further reduce size of the LES region, work should explore more realistic boundary conditions and seek to introduce free stream turbulence via SEM.

Finally, it is worth remarking that in general, the impact of decreasing the cell size on LB50 quite minimal away from the wall. This reduction in lattice number by a factor of 8 could lead to significant gains in time, particularly where much larger spatial regions are to be simulated. However, in the vicinity of the building itself, these differences are greater, which suggests that with coarser meshes, the need for wall-modelling will become more important.

## 7. Software and Computational Resources

The NS solver for the IDDES and RANS simulations is the open source CFD software OpenFOAM v6 [24], The OpenFOAM Fundation, London, UK. We ran the RANS simulation using Intel Xeon 2.4 GHz CPU processors on the ARCHER supercomputer and the IDDES simulations using AMD EPYC 7742 2.25 GHz CPU processors from the ARCHER2 supercomputer, UKRI, EPCC, Cray (an HPE company) and the University of Edinburgh, Edinburgh, UK.

The LB solver used in this paper for the LB-LES simulations is the CUDA/C++ software GASCANS [26], developed by the University of Manchester, Manchester, UK. The main LB kernel is executed on GPU; the SEM instantaneous velocity is calculated on CPU and parallelised using OpenMP. The LB-LES simulations ran using Intel Xeon Gold 6130 "Skylake" 2.10 GHz processors and one Nvidia v100 GPU card with 16 GB of GPU memory in a single node of the Computational Shared Facility (CSF) at the University of Manchester, Manchester, UK. All simulations were performed in double precision.

In this work, rather than look at raw computational speed of the different solvers and cases—A comparison of which is difficult to achieve in a meaningful manner—We adjusted the computational resources such that all simulations would take a similar order of real-world time and then analysed the accuracy and computational resources required.

This decision was driven by the fact that, increasingly, the wind engineering industry has access to substantial computational resources for running CFD, either via in-house computer clusters or via cloud-based computing. In this case, the main constraint is the practical time-frame that a simulation may be left to run before results are required; with common turnarounds within an 8-h working day, or overnight. To represent this, we adjusted core-count to achieve similar computational times per unit of flow through $t_c$.

Table 3 presents the computational resources used by each simulation, the time taken to run 5 s of physical time ( 280 $t_c$). Note that run times presented here are provided to document the time and resources consumed by the simulations in this paper and, while they are expected to be indicative of the general performance for these solvers, they are not intended as the absolute measure of the raw performance for that particular software. See Bna et al. [27] for an analysis of OpenFOAM performance and scalability and Camps Santasmasas [26] for an analysis of GASCANS performance and scalability.

Motivated by the expansion of industrial CFD onto cloud compute facilities, and to provide comparable cost, we also estimated the price of running 5 physical seconds of each simulation using the Amazon AWS EC2 facilities [28], accessed at the time of writing (January 2022). The prices are the public prices for on-demand hourly cost in the EU (London) region. We selected the AWS EC2 instances based on the work by Turner et al. [29]. The AWS instance to cost the RANS and IDDES simulations is c5n.18xlarge. c5n.18xlarge instances contain 72 vCPUs, 192GB of RAM memory and are costed at $4.608 per h ($0.064 per core-h). The instances selected to cost the LB simulations are p3.2xlarge. Each p3.2xlarge contains 1 Nvidia V100 GPU card, 8 vCPUs, 61GB or RAM memory and are costed at $3.589 per h.

From the table, we observe that IDDES2 simulation present the longest computational time, followed by LB100 and RANS. IDDES1 simulation time is slightly under RANS time and LBnoSEM is considerably lower than RANS. Regarding costs, the LB-LES simulations are the cheapest, with costs under the RANS cost. The IDDES simulations are the most expensive. LB100 and IDDES2 present the same level of accuracy and computational time but IDDES2 is around 25 times more expensive. Moreover, halving the cell size (LB50) reduces the computational time and cost of LB50 to 20% of the LB100 simulation.

The SEM calculation has a significant impact in the LB100 simulation time. LBnoSEM and LB100 run on the same computational resources and use the same mesh but the LBnoSEM simulation spends 30% of the time spend by LB100; which suggests that SEM is a bottle neck in LB100.

**Table 3.** Domain size and computational resources to run 5 s of the isolated building simulations. Time and cost relative to that required for the RANS computation. Time variation on LB-LES simulations due to activated options in the solver. Estimated costs from AWS pricing calculator [28].

|  | Cells [Millions] | Resources | Time | Estimated Cost |
|---|---|---|---|---|
| **RANS** | 3.3 | 120 CPUs | 1 | 1 |
| **IDDES1** | 6.7 | 512 CPUs | 0.8 | 3.4 |
| **IDDES2** | 19.1 | 1280 CPUs | 1.4 | 14.8 |
| **LBnoSEM** | 17.5 | 1 CPU, 1 GPU | 0.3–0.4 | 0.14–0.2 |
| **LB100** | 17.5 | 8 CPUs, 1 GPU | 0.9–1.2 | 0.42–0.59 |
| **LB50** | 2.2 | 8 CPUs, 1 GPU | 0.2–0.3 | 0.08–0.11 |

## 8. Conclusions

We presented a BGK lattice Boltzmann LES solver (LB-LES) and a WMLES (IDDES) solver nested inside a pre-calculated RANS simulation, using the RANS results as boundary conditions for the inlet, top and sides boundaries. The RANS turbulence information is input to the inlet of the LB-LES and WMLES domains via a version of the Synthetic Eddy Method, with some optimisation for GPU computing. The LB-LES solver uses GPU acceleration, while the WMLES solver is implemented on CPU using MPI parallelisation.

The results are validated using the wind tunnel experiment by Meng and Hibi [15], which models wind around a single building at $Re_H$ = 47,893.

The nested LB-LES and WMLES simulations are able to reproduce the flow characteristics and numerical values measured by Meng and Hibi [15]. Starting from the RANS solution and designing the LB-LES simulation and WMLES simulation to take similar computational time, the most accurate LB-LES simulation is around 25 times less expensive than the most accurate WMLES simulation. Moreover, increasing the cell size of the LB-LES simulation still yields comparable results and reduces the computational cost to a 30% of the the most accurate LB-LES cost. This reduction in computational resources shows the potential to run SEM LB-LES for industry relevant cases using consumer level/cloud based computational resources.

The WMLES simulation, using a regular mesh with the same resolution as LB-LES plus additional prism layer cells attached to the solid walls (IDDES2), is able to reproduce the experimental results with the same accuracy as the corresponding LB-LES simulation. The WMLES with local octree-based mesh refinement around the building (IDDES1) reproduces the correct mean velocity patterns but not the turbulent kinetic energy levels, most likely due to the dissipation of inlet turbulence by the relatively coarse mesh upstream of the building on IDDES2. This highlights the need for fine meshes upstream of the region of interest in the practical use of WMLES with inlet turbulence.

The use synthetic turbulence at the inlet, based on mean inflow data from the precursor RANS calculation, enables the LB-LES model to capture the physics of the flow correctly and present good agreement with the experimental results. Not including inlet turbulence in this way leads to an under prediction of the background turbulence levels, an under prediction of turbulent mixing around the building, and thus an over prediction in size of the downstream recirculation region. A lack of upstream turbulence was observed to have a detrimental effect on the results that is more noticeable as the domain size of the embedded LB-LES is decreased. Finally, the cell size near the building for LB-LES has a significant effect on the vicinity of the building, which suggests that where low resolution LBM is mandated, near-wall modelling will play an important role.

This study is intended primarily as a proof-of-concept, to demonstrate the potential of this framework. The test case presented in this paper is of a modest scale compared to the multiple building domains commonly studied in computational wind engineering projects. We are currently working towards the application of this framework to a larger urban region, comprising multiple buildings and much larger mesh/domain size; which will give more insight in the potential for this approach to be used in industry, as well as a way to assess computational scalability of this framework.

A more general definition for the SEM length scale limiters, $\sigma_{min}$ and $\sigma_{max}$, and a more physically realistic treatment of the top boundary condition for the nested region are still needed, and require further investigation. Despite impressive cost benefits already observed, the computational cost of the LB-LES SEM simulation might be further reduced by optimising the SEM algorithm and implementing it fully on the GPU side of the LB-LES solver.

**Author Contributions:** The contributions of each author to this paper are as follows. Conceptualisation, M.C.S. and A.R.; data curation, M.C.S. and X.Z.; LB simulations configured and run by M.C.S.; RANS and nested WMLES simulations configured and run by X.Z.; Validation, M.C.S. and X.Z.; Visualisation, M.C.S.; writing: original draft preparation, M.C.S.; writing: review and editing A.R., B.P., G.F.L.-S., J.M. and X.Z.; supervision A.R. and B.P.; Funding acquisition, A.R. All authors have read and agreed to the published version of the manuscript.

**Funding:** This research was funded by EPSRC IAA Proof of Concept Scheme grant number (EP/R511626/1/IAA273)—'Development of a dual-solver approach for fast urban wind analysis' in collaboration with WSP. The authors would like to thank EPSRC for the computational time made available on the UK supercomputing facility ARCHER and ARCHER2 via the UK Turbulence Consortium (EP/R029326/1).

**Institutional Review Board Statement:** Not applicable.

**Informed Consent Statement:** Not applicable.

**Data Availability Statement:** Publicly available datasets were analyzed in this study. The NS DNS channel flow results by Kozuka et al. can be found here: https://www.rs.tus.ac.jp/t2lab/db/ (accessed on 25 December 2021). Meng and Hibi experimental data can be found here, case A: https://www.aij.or.jp/jpn/publish/cfdguide/index_e.htm (accessed on 25 December 2021). The RANS, LB-LES and WMLES results are available on request from the corresponding author.

**Acknowledgments:** The authors would like to acknowledge the assistance given by Research IT and the use of the Computational Shared Facility at the University of Manchester.

**Conflicts of Interest:** The authors declared no potential conflict of interest with respect to the research, authorship and/or publication of this article.

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
