# Peer review of "Comparison of Lattice Boltzmann and Navier-Stokes for Zonal Turbulence Simulation of Urban Wind Flows"

_fluids, doi:10.3390/fluids7060181_

Round 1

Reviewer 1 Report

The paper deals with a method to speed up modeling of turbulent flows around buildings. Since the whole domain need to be significantly larger than the immediate neighborhood of the considered buildings the computational requirements may be very high. The Authors proposed to nest an inner LES domain into outer domain calculated using RANS methods. The RANS calculations provide boundary conditions for LES sub-domain. In such a way a trade-off between accuracy and reasonable computational demands was obtained. The work is interesting and the outcomes may be valuable for those who face the issues of modeling the large scale flows. I’ve just a few remarks listed below:

  1. Introduction section. Please state clearly what is the scientific contribution of your work. Please add a brief agenda of your paper as well.
  2. Line 79. A typo ‘hypthesize’.
  3. Please discuss possible limitations introduced by the requirement that the cells must be cubic and of constant size.

Author Response

Comment 1: Introduction section. Please state clearly what is the scientific contribution of your work. Please add a brief agenda of your paper as well.

We have added two paragraphs at the end of the introduction stating the scientific contribution of our work and the agenda of the paper.

Comment 2: Line 79. A typo ‘hypthesize’.

Typo fixed. 

Comment 3: Please discuss possible limitations introduced by the requirement that the cells must be cubic and of constant size.

Added a more detailed explanation of the effects of constant cell size to section 2.1 Lattice units. 

Reviewer 2 Report

The authors present numerical methods to interface high-resolution simulations, e.g. LES, with course mesh methods, e.g. RANS. The study is useful for several applications and well presented.

Scalability is always a question performing large simulations. I would be nice to show a the scalability of the presented methods.

In the same line, it would be nice to show a large scale application. Meshes of 20 million cells are rather small for such applications. A demonstration case with a larger mesh size would be good to include.

Author Response

Comment: 

The authors present numerical methods to interface high-resolution simulations, e.g. LES, with course mesh methods, e.g. RANS. The study is useful for several applications and well presented.

Scalability is always a question performing large simulations. I would be nice to show a the scalability of the presented methods.

In the same line, it would be nice to show a large scale application. Meshes of 20 million cells are rather small for such applications. A demonstration case with a larger mesh size would be good to include.

Scalability and performance tests for OpenFOAM and GASCANS are shown in references 25 and 26 and we aim to test the scalability of the framework presented in this paper in the near future. We are working on larger scale cases but the results are not yet ready to be included in this paper.

We've added a paragraph in the conclusions section (paragraph before last) clarifying these points.